# Neurodevelopmental milestones and associated behaviours are similar among healthy children across diverse geographical locations

José Villar[1,2], Michelle Fernandes[1,3], Manorama Purwar[4], Eleonora Staines-Urias[1], Paola Di Nicola[5], Leila Cheikh Ismail[6], Roseline Ochieng[7], Fernando Barros[8], Elaine Albernaz[9], Cesar Victora[10], Naina Kunnawar[4], Sophie Temple[1], Francesca Giuliani[5], Tamsin Sandells[1], Maria Carvalho[7], Eric Ohuma[1,11], Yasmin Jaffer[12], Alison Noble[13], Michael Gravett[14], Ruyan Pang[15], Ann Lambert[1], Enrico Bertino[16], Aris Papageorghiou[1,2], Cutberto Garza[17], Alan Stein[18], Zulfiqar Bhutta[19] & Stephen Kennedy[1,2]

It is unclear whether early child development is, like skeletal growth, similar across diverse regions with adequate health and nutrition. We prospectively assessed 1307 healthy, well-nourished 2-year-old children of educated mothers, enrolled in early pregnancy from urban areas without major socioeconomic or environmental constraints, in Brazil, India, Italy, Kenya and UK. We used a specially developed psychometric tool, WHO motor milestones and visual tests. Similarities across sites were measured using variance components analysis and standardised site differences (SSD). In 14 of the 16 domains, the percentage of total variance explained by between-site differences ranged from 1.3% (cognitive score) to 9.2% (behaviour score). Of the 80 SSD comparisons, only six were >±0.50 units of the pooled SD for the corresponding item. The sequence and timing of attainment of neurodevelopmental milestones and associated behaviours in early childhood are, therefore, likely innate and universal, as long as nutritional and health needs are met.

[1] Nuffield Department of Women's & Reproductive Health, University of Oxford, Oxford OX3 9DU, UK. [2] Oxford Maternal & Perinatal Health Institute, Green Templeton College, University of Oxford, Oxford OX2 6HG, UK. [3] Department of Paediatrics, University Hospital Southampton, Southampton SO16 6YD, UK. [4] Nagpur INTERGROWTH-21st Research Centre, Ketkar Hospital, Nagpur 440012 Maharashtra, India. [5] Ospedale Infantile Regina Margherita-Sant'Anna Citta della Salute e della Scienza di Torino, Torino 10126, Italy. [6] College of Health Sciences, University of Sharjah, Sharjah, United Arab Emirates. [7] Faculty of Health Sciences, Aga Khan University, Nairobi, Kenya. [8] Programa de Pós-Graduação em Saúde e Comportamento, Universidade Católica de Pelotas, Pelotas 96015-560, Brazil. [9] Faculty of Medicine, Universidade Federal de Pelotas, Pelotas 96015-560, Brazil. [10] Programa de Pós-Graduação em Epidemiologia, Universidade Federal de Pelotas, Pelotas 96010-610, Brazil. [11] Centre for Statistics in Medicine, Nuffield Department of Orthopaedics, Rheumatology & Musculoskeletal Sciences, University of Oxford, Oxford OX3 7LD, UK. [12] Department of Family & Community Health, Ministry of Health, Muscat, Sultanate of Oman. [13] Department of Engineering Science, University of Oxford, Oxford OX1 3PJ, UK. [14] Global Alliance to Prevent Prematurity and Stillbirth (GAPPS), Seattle Children's, Seattle 98105 WA, USA. [15] School of Public Health, Peking University, Beijing 100191, China. [16] Dipartimento di Scienze Pediatriche e dell' Adolescenza, SCDU Neonatologia, Universita di Torino, Torino 10126, Italy. [17] Johns Hopkins Bloomberg School of Public Health, Baltimore, MD 21205, USA. [18] Department of Psychiatry, Warneford Hospital, University of Oxford, Oxford OX3 7JX, UK. [19] Center for Global Child Health, Hospital for Sick Children, Toronto M5G 2L3, Canada. These authors contributed equally: Alan Stein, Zulfiqar Bhutta, Stephen Kennedy. Correspondence and requests for materials should be addressed to J.V. (email: jose.villar@wrh.ox.ac.uk)

The INTERGROWTH-21st Project consisted of several interrelated studies with the primary aim of assessing growth, health, nutrition and neurodevelopment from less than 14 weeks' gestation to 2 years of age, using the same conceptual framework as the WHO Multicentre Growth Reference Study[1]. Eight study populations across diverse geographically delimited areas were selected according to criteria for adequate health, nutrition and socioeconomic status required for constructing international standards[2,3]. The Project has produced prescriptive, international standards of fetal growth[4], newborn size and body composition[5,6], and postnatal growth of preterm infants[7]. The standards complement the existing WHO Child Growth Standards[8] very well and clearly demonstrate that participants in the INTERGROWTH-21st Project were appropriately selected to meet the WHO prescriptive criteria for optimal human growth and neurodevelopment[9].

The studies provided robust evidence of the similarities in skeletal growth from early pregnancy to 2 years of age, based on WHO recommended strategies[10]. However, the biological concept of similarities across non-isolated populations should be extended beyond skeletal growth[9], to include markers of neurodevelopment. This is justified because cognition, attentional problems, language, motor and visual capacity are fundamental human functions, and their development reflects the fast growth and maturation of the brain[11], which accompany the rapid skeletal growth that occurs in fetuses and infants[12].

The first step in our research programme was to develop, validate and test in different settings a simplified, rapid, neurodevelopment assessment package that would measure key dimensions of neurodevelopment and associated behaviours separately and objectively, and that could be used internationally across different cultures and administered to large populations[13–15].

Here, we report the findings of the second step in the programme. Using the INTERGROWTH-21st Neurodevelopment Assessment Package which we developed, we demonstrate, for the first time in a longitudinal study from pregnancy to childhood, that the sequence and timing of attainment of key neurodevelopmental milestones and associated behaviours among 2-year-old children are similar across geographically delimited populations specially selected because of their adequate health, medical care, education, and nutrition. We also show that the data from the five study sites can be pooled to create international standards because the variation in child development "between" sites is minimal compared to other sources of variance.

## Results

**Study participants.** There were 2898 singleton fetuses, born alive without congenital malformations, that were included in the construction of the international INTERGROWTH-21st Fetal Growth[4] and Preterm Postnatal Growth Standards[7], from five study sites located in the cities of Pelotas (Brazil); Turin (Italy); Oxford (UK); the central area of Nagpur (India) and the Parklands suburb of Nairobi (Kenya). All were urban regions (e.g. a complete city, or county, or part of a city with clear political or geographical limits where most deliveries occurred in health institutions). The selection criteria at the cluster level were: the areas had to be located at an altitude <1600 m, with a low risk of fetal and infant growth and developmental disturbances as well as an absence or low levels of major, known, non-microbiological contamination such as pollution, domestic smoke, radiation or any other toxic substances. Within each urban area, we selected all institutions classified locally as "private" or "corporation" hospitals and/or serving the middle to upper socioeconomic population provided that most of the institutional deliveries from the target population took place there. Women receiving

antenatal care had to plan to deliver in these institutions or in a similar hospital located in the same geographical area. Three original sites located in China, Oman and the USA did not participate in the sub-studies that are the focus of the present report.

One hundred and ninety women were lost to follow-up or withdrew consent soon after pregnancy or their neonates died before hospital discharge. Although this was an a priori designed follow-up study, there was delay in obtaining the required funding. Hence, by the time the follow-up started, 833 children were considered unlikely to be evaluated within the assessment age-window, as per protocol. Additionally, 117 parents withdrew consent.

From the 1758 remaining children, five died during infancy and 83 had a clinical evaluation but not the complete neurodevelopment assessment (Fig. 1). We also could not contact 331 parents. Thus, of the 1753 eligible children, 1422 (81.1%) were seen at 2 years and 1339 (76.4% of all those eligible) had a neurodevelopment assessment. Among the latter, 32 children were diagnosed with severe visual problems, seizures, hearing impairment, malaria or heart conditions. The diagnoses were made as part of routine clinical care by the appropriate clinicians; consistency across sites was maintained by providing a manual of operations to standardise care, which is freely available at www.intergrowth21.org.uk. These children were excluded from the final analysis, leaving data from 1307 available children (683 girls and 624 boys) (Fig. 1).

The proportional contribution to the analysis sample from the study sites was 15% Brazil ($N = 199$), 24% India ($N = 318$), 24% Italy ($N = 311$), 24% Kenya ($N = 311$) and 13% the UK ($N = 168$). The baseline socio-demographic characteristics (Supplementary Table 1) of these samples were remarkably similar to those of the original cohorts at each study site[2], and the samples across the five sites were, as expected, also similar because the same entry criteria were used. In addition, the satisfactory growth and development of the children at 1 and 2 years of age has confirmed the adequate health, educational and nutritional status of the original cohorts[9].

The median, inter-quartile range (IQR) age at the neurodevelopment assessment was 24.2 (IQR 23.9, 25.2) months; 1128 (86.4%) children were evaluated between 22 and 26 months. The neonatal characteristics of the children evaluated at 2 years ($n = 1307$) were compared to those of the children lost to follow-up ($n = 331$). The two samples were very similar in terms of anthropometric measures at birth, preterm birth rate, sex and neonatal morbidity. At hospital discharge, over 90% of newborns were exclusively breastfed in both groups (Table 1).

Exclusive/predominant breastfeeding was discontinued at a median of 5.5 (IQR 4.0, 6.0) months in the evaluated group and at 5.0 (IQR 3.0, 6.0) months in the non-evaluated group. Breastfeeding stopped entirely at a median of 12.0 (IQR 7.0, 18.0) months in the evaluated group. The median years of maternal education in the evaluated group were 15.0 (IQR 13.0, 17.0) and 15.0 (IQR 12.0,16.0) in the non-evaluated group.

The majority of infants were vaccinated in accordance with recommended policies and only 16.5% were treated with iron/vitamins. The most frequently reported or diagnosed conditions during the second year of life were exanthema/skin diseases, fever ≥ 3 days during ≥3 episodes, and otitis media/lower-tract respiratory infections. Antibiotics were prescribed to 12.5% of children during their second year. Overall, the children evaluated had low morbidity in accordance with their mothers' low-risk status, the very low preterm birth rate (4%) and the families' socioeconomic and educational status (Tables 1 and 2).

By 2 years, 92% of length, 90% of weight and 91% of head circumference measures of this cohort's values were within the

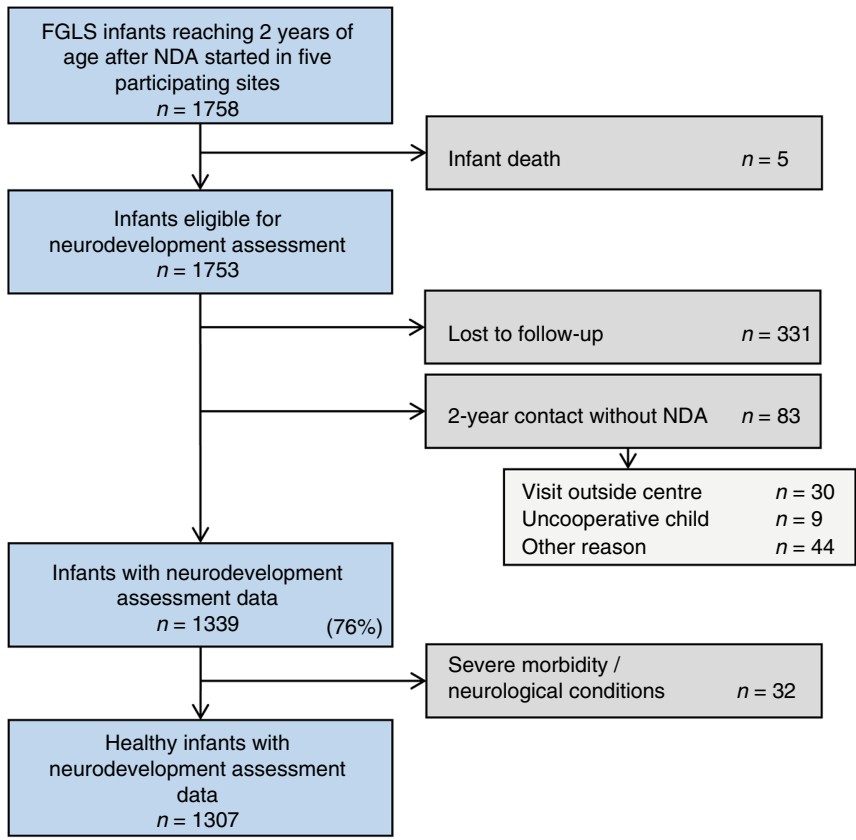

**Fig. 1** Participant flow: INTERGROWTH-21st Project Neurodevelopment Assessment cohort at 2 years of age

| Table 1 Neonatal characteristics of children included in the INTERGROWTH-21st Neurodevelopment Assessment Study compared to children lost to follow-up | | |
|---|---|---|
| | Children evaluated (n = 1307) | Children lost to follow-up (n = 331) |
| Gestational age at delivery, weeks | 39.4 (1.5) | 39.3 (1.5) |
| Birthweight, kg | 3.2 (0.5) | 3.2 (0.5) |
| Birth length, cm | 49.0 (2.0) | 49.0 (2.1) |
| Head circumference at birth, cm | 33.9 (1.4) | 34.0 (1.3) |
| Apgar at 5 min | 9.5 (0.6) | 9.6 (0.7) |
| Age at hospital discharge, days[a] | 3.0 (2.0, 4.0) | 2.0 (1.0, 3.0) |
| Boys | 624 (47.7) | 160 (48.3) |
| Preterm birth (<37 weeks' gestation by LMP) | 54 (4.1) | 16 (4.8) |
| Early preterm (<34 weeks' gestation by LMP) | 6 (0.5) | 2 (0.6) |
| NICU stay > 1 day; <3 days | 47 (3.6) | 18 (5.5) |
| NICU > 3 days | 33 (2.5) | 12 (3.6) |
| Hyperbilirubinaemia | 66 (5.1) | 18 (5.5) |
| Respiratory distress syndrome | 29 (2.2) | 7 (2.1) |
| Transient tachypnoea of the newborn | 18 (1.4) | 12 (3.6) |
| Exclusive breastfeeding at hospital discharge | 1209 (92.6) | 300 (90.9) |

Data are means (SD) or proportions (%) unless otherwise specified. Missing data below 2% for all variables
*LMP* last menstrual period, *NICU* neonatal intensive care unit
[a]Median (inter-quartile range)

3rd and 97th centiles of the WHO Child Growth Standards. Among infants born at term, the mean (SD) age- and sex-specific z-scores for length, weight and head circumference were −0.15 (1.00), 0.23 (1.05) and 0.13 (1.08), respectively.

**Developmental measures**. The number of observations available across sites for individual domains of the INTERGROWTH-21st Neurodevelopment Assessment, Cardiff tests and age of achievement of two WHO gross motor development milestones are shown in Supplementary Table 2. Ninety-nine percent of children reached the milestone of the age at "first standing alone" and 98% reached the milestone of the age at "walking alone" within the 3rd and 97th centiles of the WHO age-window of achievement[16]. Furthermore, 99% of children had binocular visual acuity of 0.4 log Mar (6/15 Snellen equivalent) or better, and 98.4% had visual contrast sensitivity of 33.3 or higher suggesting that they did not have any visual impairment[17].

The mean ages at assessment were almost identical for girls and boys (25.1; SD 2.2 and 25.0; SD 2.1 months, respectively). There

was a trend towards higher average values in the cognitive and motor scores among girls, which was inverted for attention and language scores in favour of boys but without a clear overall differential pattern. Nevertheless, we have adjusted all comparisons across study sites by sex. As expected, the age of the children (in months) at the time of assessment had an effect, i.e. older children, even within the assessment age-window, had slightly higher scores. We have, therefore, adjusted all comparisons across study sites for chronological age at assessment.

Table 3 presents the percentage of the total variance that can be attributed to differences between study sites (as opposed to differences among individuals within sites) estimated using variance components analysis in these five healthy, adequately nourished cohorts.

Amongst the primary domains (top seven domains in Table 3), the proportion of the total variance adjusted by sex and age at assessment explained by site differences was 1.3% for the mean cognitive score and 5.4% for the mean attentional problems score; for the visual tests and motor domains, site differences explained 8% or less of the total variance for each item. Results from the models that included sex and age were consistently similar to those that included fetal and child growth variables.

Furthermore, we selected an item from the cognitive domain that could be considered as describing one of the three components of executive function[18], i.e. "inhibition on task performance" ("Identifies glitter bracelet under washcloth"). Differences between sites explained only 3.4% of the total variance for this item (Table 3). This is relevant because executive function describes a set of skills involved in planning complex behaviours towards the purpose of solving novel problems[19].

Similar estimations were made for the seven secondary domains (Table 3). For positive affect (mood), receptive and expressive language and overall behaviour, differences between sites explained between 1.9 and 9.2% of the total variance. The emotional reactivity (14.2%) and negative behaviour (14.1%) domains were the only two in which site differences explained a relatively higher proportion of the total variance (Table 3).

Finally, the same analyses were performed using reported "age at first standing" and "walking alone", both fundamental motor development milestones. The percentage of the total variance explained by between-site differences was 5.6 and 6.9%, respectively (last two items of Table 3).

As the second strategy to evaluate similarities between study sites, the SSD for each of the 16 developmental domains was estimated for the five sites. The means and SDs were adjusted by sex and age at assessment, expressed as units of the SD of all sites combined (Fig. 2). For the seven primary domains complemented by the two WHO gross motor milestones, there were only two

**Table 2 Morbidity diagnoses between 1 and 2 years of life of children included in the INTERGROWTH-21st Neurodevelopment Assessment Study**

|  | Children in the analysis (n = 1307)[a] |
|---|---|
| Hospitalised at least once | 117 (9.0) |
| Any prescription made by a health-care practitioner: | 771 (59.0) |
| • Antibiotics (≥3 regimens) | 163 (12.5) |
| • Iron/folic acid/vitamin B12/other vitamins (≥3 regimens) | 216 (16.5) |
| Up-to-date with local vaccination policies | 1231 (94.3) |
| Otitis media/Pneumonia/Bronchiolitis | 98 (7.5) |
| Parasitosis/Diarrhoea/Vomiting | 51 (3.9) |
| Exanthema/skin disease | 159 (12.2) |
| Urinary tract infection/pyelonephritis | 6 (0.5) |
| Fever ≥ 3 days (≥3 episodes) | 147 (11.3) |
| Other infections requiring antibiotics | 44 (3.4) |
| Asthma | 15 (1.1) |
| Gastro-oesophageal reflux | 4 (0.3) |
| Cow's milk protein allergy | 11 (0.8) |
| Food allergies | 15 (1.1) |
| Injury trauma | 34 (2.6) |
| Surgery | 9 (0.7) |

Data are number (%). Missing data below 1% for all variables
[a]For five children, information on morbidities in the first year of life was used

**Table 3 Variance components analysis for individual domains of the INTERGROWTH-21st Neurodevelopment Assessment Package showing variance between study sites as % of the total variance**

| Neurodevelopment domain | Model with age and sex (n = 1307) | Model with age, sex and fetal HC (n = 1302) | Model with age, sex, HC and length at 2 years (n = 1283) |
|---|---|---|---|
| Cognitive score (%)[a] | 1.3 | 1.2 | 1.2 |
| Executive function-like (%)[a] | 3.4 | 3.3 | 3.3 |
| Attentional problems score (%)[b] | 5.4 | 5.8 | 4.1 |
| Visual acuity (%)[a] | 8.0 | 6.9 | 7.4 |
| Visual contrast sensitivity (%)[a] | 7.3 | 7.3 | 6.9 |
| Fine motor score (%)[a] | 6.0 | 6.0 | 6.2 |
| Gross motor score (%)[b,c] | 6.5 | 7.2 | 6.7 |
| Receptive language score (%)[b,c] | 1.9 | 1.4 | 1.9 |
| Expressive language score (%)[b,c] | 7.3 | 7.0 | 7.3 |
| Positive behaviour score (%)[c] | 8.5 | 8.2 | 8.7 |
| Negative behaviour score (%)[c] | 14.1 | 13.6 | 13.7 |
| Total behaviour score (%)[c] | 9.2 | 9.2 | 9.9 |
| Emotional reactivity score (%)[b] | 14.2 | 15.3 | 14.0 |
| Positive affect score (%)[b] | 2.5 | 2.7 | 2.7 |
| Age when first stood alone (%)[b] | 5.6 | 5.3 | 6.5 |
| Age when first walked alone (%)[b] | 6.9 | 6.9 | 7.6 |

Variance components estimated by nonparametric generalised linear mixed models including covariates as fixed effects and study site as random effect
*Age* age at assessment, *fetal HC* ultrasound head circumference z-score from 26 to 34 weeks' gestation compared to the INTERGROWTH-21st Fetal Growth Standards[2]; Head circumference and length z-scores at 2 years compared to the WHO Child Growth Standards[8]
[a]Direct administration
[b]Caregiver report
[c]Concurrent observation

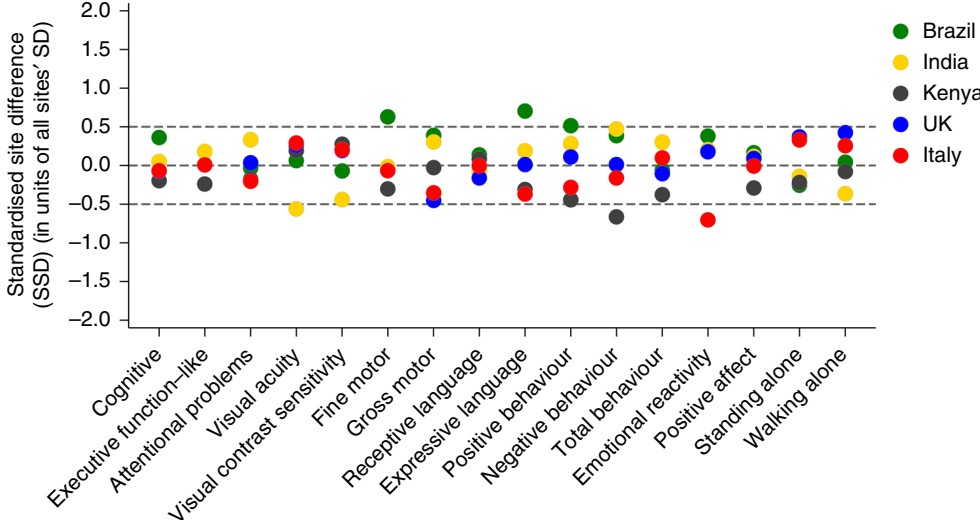

**Fig. 2** Standardised site difference for mean scores of individual domains of the INTERGROWTH-21st Project Neurodevelopment Assessment, Cardiff tests and age of achievement of two WHO gross motor development milestones. The standardised site differences calculated as the site mean of each domain minus the mean for all sites combined divided by the SD of all sites together, adjusted by the age at the time of assessment and sex. Results are shown as: Brazil (green), India (yellow), Kenya (brown), UK (blue) and Italy (red). Sample size according to domain and study site is given in Supplementary Table 2

values of the 45 comparisons outside the pre-specified SSD cut-off point of ±0.5 (Brazilian site for fine motor: SSD 0.63 and Indian site for visual acuity: SSD −0.56).

Considering the seven secondary NDA domains (35 comparisons), there were only three SSD values outside the pre-specified range (Brazilian site for expressive language: SSD 0.68, Kenyan site for negative behaviour: SSD −0.66, and Italian site for emotional reactivity: SSD −0.70) (Fig. 2).

## Discussion

We have presented evidence that healthy, adequately nourished, well-educated pregnant women, recruited from five diverse geographical and cultural study sites, who receive recommended antenatal care, have children that display consistent similarities at 2 years of age across a comprehensive set of neurodevelopmental outcomes. The evidence complements previous reports from the INTERGROWTH-21st Project demonstrating equivalent similarities across these study sites for skeletal growth from the first trimester of pregnancy to 2 years of age[9].

In 14 of the 16 domains evaluated, the percentage of variance explained by between-site differences ranged from 1.3% (cognitive score) to 9.2% (behaviour score) of the total variance. Of the 80 comparisons using SSDs, only six were >±0.50 units of the pooled SD for the corresponding item, two of them just marginally outside that limit and without any specific pattern. The percentage of variance explained by site differences in the emotional reactivity and negative behaviour domains (14%) could be because the scoring of these items is more culturally dependent, i.e. negative behaviour could be perceived and hence scored differently across settings[20].

It is evident that across developmental and growth parameters, only a very small percentage (around 10%) of the total variance in these fundamental human functions can be explained by differences among these populations (Fig. 3). The present results and previous publications, presented together in Fig. 3, support the position that most of the observed differences in growth and neurodevelopment across general populations or countries are primarily due to socioeconomic, educational and class disparities, i.e. postal codes define the health profiles of humans better than their genetic code[21].

Our study has some unique features. It is based on a set of multi-site, prospective cohorts that consisted of healthy, adequately nourished, well-educated mothers and their babies, enrolled to test a specific hypothesis, that were followed from early pregnancy to 2 years of age. Anthropometric, visual and gross motor development scores agreed well within recommended limits for normal populations. Detailed information was obtained about the socioeconomic, education and environmental backgrounds of the families selected at both population and individual levels[3]. All clinical, anthropometric measures, feeding practices and monitoring procedures were rigorously standardised across time and sites. The samples described are not representative of whole countries, regions or cities, nor were they intended to be. Instead, they were specifically selected—reflecting our conceptual approach to the issue—to include geographical areas populated by low-risk pregnant women and their children within each country or city. Hence, the samples are intended to represent a theoretical, healthy, low-risk population—rather than one that includes both low- and high-risk populations—within a country or city.

We initially considered other short assessment tools during the preparation of the study[14]. While some employ a global approach, our study design required a tool based on a multi-country sample[9] that could separately evaluate a range of domains. Moreover, we were interested in combining psychometric methodologies, i.e. the direct administration of tasks, concurrent observation of child performance and caregiver reports, to balance the risk of recall and reporter bias[13]. Our efforts at scrutinising the pre-existing literature showed that such a tool was not available[14]. Hence, we developed, standardised and validated a new, domain-focused, culturally neutral, and simple-to-implement assessment tool[13,14]. For maternally reported items on attentional problems and emotional reactivity, local language versions from validated Child Behaviour Checklist translations were produced[22].

We a priori selected a set of seven primary domains and within them, the cognitive domain of the INTER-NDA was considered the primary outcome because its constituent items are directly administered to the child in task-based sequences; it is less affected by cultural factors, and it is scored objectively so it is not affected by recall or parental report bias. The cognitive domain is also strongly associated with adverse pregnancy outcomes such as impaired fetal growth[23].

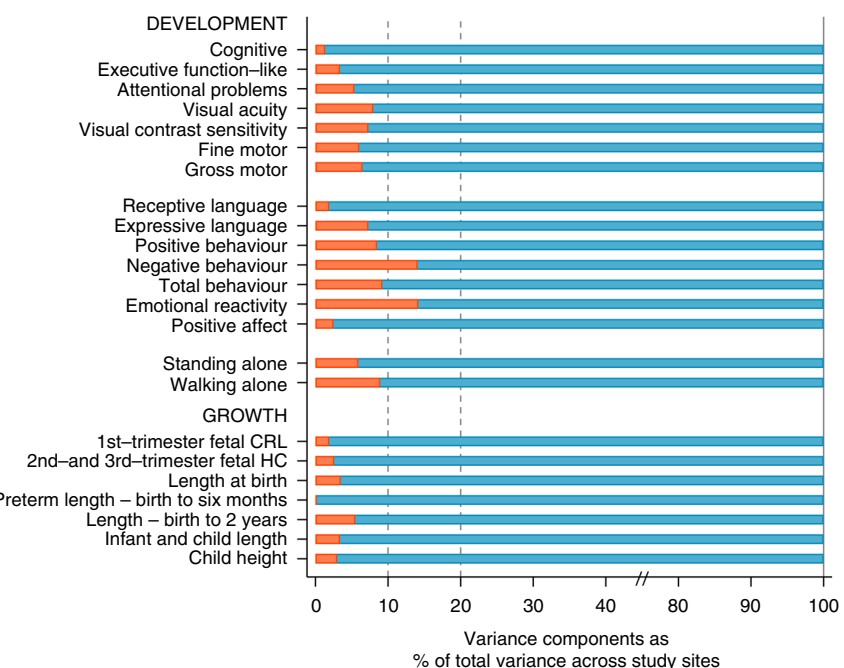

**Fig. 3** Variance components analysis of 16 neurodevelopmental domains evaluated in the present study (upper part) and variance components analysis of 7 measures of fetal, newborn, infant and child growth (lower part). Red bars are the % of total variance explained by between sites variability for each domain or growth measure. Data for the seven growth measures: INTERGROWTH-21st Project (eight study sites)[2,3,5,9]; WHO Multicentre Growth Reference Study (five study sites)[8]; Habicht et al.[41]

Interestingly, the two language domains were considered a priori secondary outcomes for comparing the cohorts because of their strong association (particularly expressive language) with children's interactions with their cultural environment and the influence of care providers. However, our observation of similarities in the children's performance on the language domains suggests that, overall, children may not have differed dramatically in the levels of early stimulation they received from these urban, well-educated families (median of 15 years of maternal formal education) (http://data.uis.unesco.org/Index.aspx?queryid=242).

Alternatively, the findings could support the concept that language acquisition is mostly a function of cognitive maturation[24,25], affective and social connectedness to other persons (universal to all cultural contexts) and, as such, children with high levels of cognitive and social functioning would, in most cases, display similarly high levels of language skills[26,27].

We selected 2 years of age as the time-point for the key development assessment of the entire study because growth markers at this age have been found to be predictive of intelligence, school performance, adult nutrition and human capital in high-, middle- and low-income settings[28–32]. Moreover, this is the earliest age at which the assessment of development is not confounded by transient neurological syndromes of prematurity, and at which conventionally used developmental instruments, such as the Bayley Scales of Infant Development, have been found to possess an acceptable level of medium- and long-term predictive validity[13,33]; this age also corresponds to the end of Piaget's sensorimotor stage[34].

The psychometric properties of the measure we developed, the INTER-NDA, including internal consistency and construct validity, evaluation of its performance using interclass correlations for absolute agreement; Bland—Altman analyses for bias and limits of agreement, and sensitivity and specificity analyses for accuracy, have all been validated against the Bayley Scales of Infant Development[14,15].

In our statistical analyses, we adjusted by the age of the child at the developmental assessment, which controls for self-selection of families attending follow-up clinics, i.e. health-care-seeking patterns. We did not adjust for any characteristics at recruitment early in pregnancy as the main hypothesis of the paper is to evaluate the differences among the study samples that were selected using the same entry criteria. We opted for this conservative approach because further adjustment for any "residual confounding" would have made the samples more "artificially" similar.

We successfully evaluated 76% of the eligible children, despite the known difficulties in retaining healthy participants in large-scale, multi-national, follow-up studies. Selection bias is unlikely in view of the baseline similarities between the children evaluated and those lost to follow-up. Unfortunately, we could not include three of the original sites that participated in the INTERGROWTH-21st Project because initiation of the follow-up study was delayed due to the funding process. This reduced the external validity of our observations although we retained five study sites on four continents, with distinct geographical and cultural identities. It also affected the total size of the original cohort; however, this is less relevant to our hypothesis because the comparisons were focused on the sites that contributed data, which retained a large proportion of their original sample.

An additional limitation of the analysis is that we did not have repeated developmental measures: hence, while we were able to estimate the percentage of variance explained by "between sites' variation", i.e. the main hypothesis of this study, it would have been informative to estimate the "within site variation" but this requires individual repeated measures at different ages. There are considerable conceptual difficulties in identifying comparable repeated measures of developmental domains at both 1 and 2 years of age. We have, however, recently published results comparing the percentage of "within" versus "between" sites variance for repeated skeletal growth measures, i.e. infant length for the same cohort[2]. These data for the "between" variance component

(5.5% of the total variance for length), that is considerably lower than the "within" variance (42.9% of the total variance for length), are very similar to the results presented here.

An inevitable constraint, inherent in the brief psychometric tools required to evaluate large populations, is that some domains are based on only a few items. This was the case in our study for negative behaviour, positive affect and receptive language, which were evaluated by two items. Therefore, we concentrated our interpretation and conclusions on the consistent similarities observed overall.

Traditionally, developmental comparisons during childhood across populations have been made between developing and developed countries; low and high socioeconomic populations; children of parents with high and medium to low levels of education, or immigrant populations with native children in the USA or Europe[35–38]. When attempts were made across contexts to compare levels of parental stimulation, which is key to early child development (ECD), samples have come from very different socioeconomic contexts and likely health conditions[39,40]. These studies suggested that cognition, language, play and sociability are heavily influenced by the socioeconomic structures of the society in which the children are growing up. Such evidence has been used to explain developmental differences between socio-economic levels in industrialised societies, where most of the research has been conducted.

We have approached this fundamental question differently, although in a complementary manner. We studied cohorts of healthy, well-educated, adequately nourished women receiving evidence-based pregnancy care from culturally different geographical areas that were selected because of their low morbidity and environmental risks. We have documented that under similar socioeconomic, health and nutrition conditions, there are remarkable similarities in the physical growth of their children up to 2 years of age[9], to which we now add evidence of comparable similarities in the attainment of neurodevelopmental functions and associated behaviours.

The initial INTERGROWTH-21st Project publications stressed that the relevant question when comparing healthy, low-risk populations to identify physical growth differences or similarities is: whether or not the variability in skeletal growth *within a population*, i.e. inter-individual difference, is larger than the variability *between populations*, i.e. inter-population difference, when nutritional, socioeconomic, environmental and health-care needs are met. There is now consistent evidence that the variability in human skeletal growth within a population is seven times larger than that between populations (genetic variability), which represents less than 10% of the total variance[2,8,41,42].

The magnitude of the results presented is entirely consistent with studies of the genetics of human growth. For example, genome-wide association studies (GWAS) testing for common fetal variant effects on birthweight have to date identified as many as 60 associated genetic loci[43], and estimation of the variance in birthweight due to a distinct maternal genetic contribution ranges from 3 to 22%[43,44]. Similarly, GWAS have identified nearly 700 independent variants within over 400 genetic loci that together explain only 20% of the heritability of adult height[45].

Interestingly, a very recent report studied a cross-sectional sample of clinically healthy, 0-to-42-month-old children using a short, pre-coded interview with caregivers to assess age of achievement of 106 developmental milestones. The study was conducted in 22 health-care clinics in four diverse low-middle income countries (LMICs) with ethnic, cultural, and language differences[46]. Using predefined criteria of practical equivalence, almost all milestones at 1 year of age and 76% of the milestones up to 3 years of age were attained at similar ages across the four study sites. Despite the considerable methodological differences

with our study, this report provides evidence from four other populations of the similarities of early developmental patterns among healthy children. The authors concluded with the statement that internationally applicable tools were needed to "assess children's development to guide policy, service delivery, and intervention research that might help narrow the gap between high-income countries and LMICs in addressing early childhood development".

We agree completely with this statement, especially in the light of the similarities in results obtained for anthropometric measures across populations when health and nutritional needs are met. International prescriptive standards for ECD are recommended for both clinical practice and research as many of the tools currently used are based on very skewed populations from developed countries, which little resemble present, culturally mixed societies. For example, personality tests, which have been used on millions of subjects worldwide, were constructed using a few hundred Swiss-German patients in the 1960s in the case of the Rorschach test, and white, rural, Protestant "Minnesotan normal" hospital visitors in the 1930s in the case of the Minnesota Multiphasic Personality Inventory[47,48].

Clearly, in clinical settings, it is important to focus on an individual child as the unit of diagnosis and interventions. At population level, the first step is to identify children at risk, i.e. those that require further assessment and are most likely to benefit from an intervention. However, the heterogeneity of ECD measures presently in use and their reliance on specialists has made it difficult to carry out population-based ECD screening.

Our results strongly support the construction of international, psychometric, ECD standards (manuscript submitted). Our strategy not only provides more variability to the distribution of scores, which is desirable for a first-level screening tool, but also better reflects the underlying constitution of modern multicultural societies.

Finally, it is worth noting that we have deliberately avoided reporting the race/ethnicity of the participants despite some widely held beliefs that it influences ECD. We have consistently argued that the use of self-reported race/ethnicity in scientific publications is problematic in most non-isolated populations because, in addition to the inherent biases of self-reporting, there is large ancestral admixture due to global migration, invasions and other population movements. Furthermore, there are at least 116 definitions of self-reported race/ethnicity in the biomedical literature[49].

In short, our neurodevelopmental and skeletal growth results from conception to childhood, as well as the genetic evidence summarised by Craig Venter in reference to possible links between race and intelligence: "There is no basis in scientific fact or in the human genetic code for the notion that skin colour will be predictive of intelligence" (https://www.theguardian.com/news/2018/mar/02/the-unwelcome-revival-of-race-science) strongly support our conclusions.

## Methods

**Study subjects.** INTERGROWTH-21st was a large, multi-centre, population-based project conducted between 2009 and 2016, in eight delimited geographical areas worldwide. The primary aim was to study growth, health, nutrition and neurodevelopment from early pregnancy to 2 years of age in populations of mothers and children with optimal health. A geographical area was a complete city or county, or part of a city with clear political or geographical limits, located at an altitude <1600 m, with low-risk health indicators for perinatal morbidity and mortality, in which women receiving antenatal care had plans to give birth within the area, that had to be free or have low levels of major, known, non-microbiological contamination[3].

The *Fetal Growth Longitudinal Study* (*FGLS*), one of the main studies of the INTERGROWTH-21st Project, recruited pregnant women from the aforementioned populations, who met the individual entry criteria of health, nutrition, education, and socioeconomic position, and accurate gestational age

estimation. The objective was to construct international standards for gestational weight gain, early and late fetal growth, newborn size and preterm postnatal growth[4,5,7,50,51]. The cohort enrolled in FGLS was followed up to 2 years of age, and evaluated for their skeletal growth, nutrition, health and the WHO gross motor milestones[9,16].

Across all study sites, standardised clinical care and feeding practices were implemented based on protocols developed by the INTERGROWTH-21st Neonatal Group (http://www.intergrowth21.org.uk). Exclusive breastfeeding up to 6 months was promoted for all babies, supplementing preterm infants as recommended[52]. Standardised information was obtained during pregnancy, at birth and at 1 and 2 years of age on health, anthropometric measures, severe morbidities, length of breastfeeding, timing of the introduction of food, feeding practices and food intake using forms specially produced for the project (www.intergrowth21.org). The baseline characteristics of the full cohort and follow-up methodology have been published recently[9].

For the neurodevelopmental evaluation, children were scheduled for assessment at 2 years of age in five of the eight original sites: the cities of Pelotas (Brazil), Turin (Italy), Oxford (UK), the central area of Nagpur (India) and the Parklands suburb of Nairobi (Kenya) using tools specifically developed or selected for this purpose[13]. All participants were part of the original FGLS cohort and contributed data towards the construction of the international Fetal Growth and the Preterm Postnatal Growth Standards[4,7]. The sites in China, Oman and the USA did not participate because of logistical and administrative reasons, delays in the start of the study and/or staff availability.

**Assessment tools**. The INTERGROWTH-21st Neurodevelopment Assessment (INTER-NDA) is a brief, objective tool, measuring multiple dimensions of early development, targeted at children aged 22–30 months[13]. It was designed to be implemented by non-specialists across international settings[14], and includes a reduced number of culture-specific items measuring cognition, expressive and receptive language, motor skills (fine and gross), positive and negative behaviour, attentional problems and social-emotional reactivity (taken from the Child Behavior Checklist)[53] using a combination of directly administered, concurrently observed and caregiver reported items. The INTER-NDA has been validated against the Bayley Scales of Infant Development III edition[47], showing good to moderate agreement[14], and has shown good levels of inter-rater ($k = 0.70$; 95% CI 0.47–0.88) and test/re-test reliability ($k = 0.79$; 95% CI 0.48–0.96)[13].

The gross motor domain of the INTER-NDA was complemented by the evaluation of the age of achievement of the matching WHO gross motor development milestones "standing alone" and "walking alone". Information was obtained at both the 1- and 2-year follow-up visits in order to evaluate consistency. In cases of disagreement (19 cases out of 1292 for "standing alone" and 18 cases out of 1296 for "walking alone"), the 1-year information was selected as the preferred data point.

Vision was assessed using the Cardiff Visual Acuity and Contrast Sensitivity tests[17] for binocular vision. These tests are indicators of the integrity of the visual pathway and central nervous system and were considered robust "biological controls", complementing the fine and gross motor items. Importantly, both tests are directly observed neurodevelopmental markers, unlikely to be affected by cultural influences.

The administration of the INTERGROWTH-21st Neurodevelopment Package was supported by an electronic and tablet-based data collection and management system developed for this study, which contains the INTER-NDA operation manual, visual cues, examples, and fully integrated quality checks[13]. Staff administering the assessments were aware of the general principles of the Project but not the specific hypothesis being tested. In addition, they were unaware of the INTER-NDA domain and total scores for individual children, as well as for their own and all the other study sites. Data were uploaded to centralised data-servers as soon as each assessment ended. The data management team performed monthly checks on the site-based and centralised databases (www.intergrowth21.org.uk/protocol.aspx?lang=1).

**Statistical analyses**. The sample studied depended on pragmatic considerations. The present report is the 2-year follow-up of the initial cohort of pregnant women studied from early pregnancy[1,8]. The total number of eligible children to be measured at 2 years of age was therefore fixed. The sample size estimations of the original cohort (approximately 500 fetuses per site) focused on the precision and accuracy of the extreme centiles of the complete population, i.e. the 3rd or 97th centile because they correspond closely to ±2SD, and they are the recommended cut-offs of the WHO Child Growth Standards, which are used internationally to evaluate children of this age; however, in the present study, such estimations do not apply because of the different nature of the hypothesis.

In this component of the study, the neurodevelopment of 1307 children was evaluated at 2 years of age, an average of 261 children per site. This sample size was considered adequate to explore the predicted small site-specific differences. Post-hoc power calculations showed that the study was sufficiently powered to observe small differences among study sites (calculations for all domains with power >0.99) and small effect sizes for the between-group variances. For example, for a between-group variance of 10% of the total variance and a two-tail alpha of 0.05, the power is 0.84.

Primary comparisons across sites were conducted using the average values of cognition, attentional problems, fine motor, and gross motor, visual acuity and contrast sensitivity scores. Expressive language, receptive language, positive, negative and total behaviour, and social-emotional reactivity were considered secondary outcomes. Members of the original INTERGROWTH-21st Neurodevelopmental Group selected these two groups of outcomes on an a priori basis, founded on the literature and relative robustness of items used to assess the primary outcomes to cultural influence.

Among the primary domains, the cognitive was also considered a priori to be the most appropriate single primary outcome because its constituent items are directly administered to the child in task-based sequences and are, therefore, unaffected by recall or parental reporter bias. Furthermore, consistent associations between prenatal/perinatal outcomes and cognitive scores at 2 years of age have been reported[54–57].

On the basis of our previous publications and WHO recommendations for comparing the similarities in growth across populations[2,4,5,8], we used variance components analysis to calculate the percentage of total variance due to between-country differences for each of the domains and the two visual tests. Variance components were calculated by nonparametric generalised linear mixed models (using the STATA package GLLAMM), which included as covariates sex, fetal head circumference obtained between 26 and 34 weeks' gestation (expressed as z-scores of the international INTERGROWTH-21st Standards), age at assessment in months and weeks and 2-year length and head circumference (expressed as z-scores of the WHO Child Growth Standards). These variables were included as fixed effects with study site as random effect. The method was chosen because it assumes a discrete instead of a Gaussian distribution of the random effects and showed better fit than Maximum Likelihood and Restricted Maximum Likelihood models according to the lower Akaike information and Bayesian information criteria.

In addition, for each of the domains and visual tests, SSD were calculated as the difference between the mean from a given site and the mean of all sites together, expressed as a proportion of the SD of the pooled data across sites for each specific domain; means and SDs were adjusted by sex and age at examination (centred on 24 months of age). As pre-specified in the INTERGROWTH-21st Project protocol and recommended by WHO[8], SSD values in the range of ±0.5 units of the pooled SD were taken as adequate to determine whether the data from all sites could be pooled[58].

For all analyses, Stata 15 software was used (StataCorp. 2017. Stata Statistical Software: Release 15. StataCorp LLC, College Station, TX). Data were entered locally into the specially developed, online data management system (http://medscinet.com).

The INTERGROWTH-21st Project was approved by the Oxfordshire Research Ethics Committee "C" (reference: 08/H0606/139), the research ethics committees of the individual institutions and the regional health authorities where the project was implemented. Written informed consent was obtained from all participants. The sponsors had no role in the study design, data collection, analysis, interpretation of the data, or writing of the paper. The following authors had access to the full raw dataset: J.V., E.S.U., E.O. and S.K. The corresponding author had full access to all the data and final responsibility for submitting the paper.

**Reporting summary**. Further information on experimental design is available in the Nature Research Reporting Summary linked to this article.

## Data availability
The data that support the findings of this study are available from the corresponding author upon reasonable request.

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

## Acknowledgements

This study was funded by the INTERGROWTH-21st grant 49038 from the Bill & Melinda Gates Foundation to the University of Oxford; we gratefully acknowledge their support. A. P. is supported by the National Institute for Health Research (NIHR) Oxford Biomedical Research Centre (BRC). M.F. is supported by an NIHR Academic Clinical Fellowship.

## Author contributions

J.V. and S.K. conceptualised and designed the INTERGROWTH-21st Project. J.V., S.K., and A.N. prepared the original protocol, with later input from A.P., L.C.I., F.B., and Z.B. J.V., A.P., L.C.I., A.L., and Z.B. supervised and coordinated the Project's overall under-taking. E.S.U. and E.O. carried out data management and analysis in collaboration with J.V. R.P., F.B., M.G., M.C., Y.J., E.B., M.P. collaborated in the overall project and

implemented it in their respective countries. P.Di.N., S.T., T.S., R.O., N.K. and E.A. performed the neurodevelopment assessments. M.F. and A.S. led the neurodevelopment assessment component. F.G. assisted in the global coordination of the project. J.V. and S. K. wrote the report with significant contributions by C.G., C.V., and F.B. All co-authors read the report and made suggestions on its content.

## Additional information

**Competing interests:** The authors declare no competing interests.

