## [Peer Review File · Nature Communications]

REVIEWERS' COMMENTS:

Reviewer #1 (Remarks to the Author):

Overall, this revised manuscript has addressed the concerns of the previous review. This reviewer continues to believe, however, that reporting ethnicity for the samples is important (even if flawed) to better understand the cohorts studied. Even if one does not believe that ethnicity has an impact on cognitive functioning with respect to genetic background, ethnicity in many situations can be a proxy for social and environmental issues that could impact cognitive development. I believe that readers of this manuscript will want that basic information. It is not necessary to relate ethnicity to the developmental outcome.

Reviewer #2 (Remarks to the Author):

I am satisfied that the authors have addressed my concerns and that this manuscript adds important information to the field.

Reviewer #3 (Remarks to the Author):

Thank you for the opportunity to re-review this paper. I have reviewed the Authors' responses to previous comments. A number of comments have been addressed but there are still some comments that require consideration by both the authors and the editors in regard to details of both recruitment and the details of the measures themselves. While these have been published elsewhere and cited it would seem reasonable that some further detail is included in this paper. The issue of the psychometrics and validity of the developmental measurement tools is particularly important given the conclusions. There still appears to be limited interpretation and discussion of the paper in the context of what it is trying to achieve and the in the context of child development and measurement more generally. The authors have mentioned some of these implications in their responses to reviewers including implication for further use of their measurement approach in the context of developmental tools in low and middle income countries and some of the current limitations but have not included sufficient responses in the paper itself. Similarly while the issue of deriving a theoretical population that is not representative makes sense and is alluded in the response comments this still needs further clarity in the paper itself as will seem odd to most reviewers. The issue of population variance (within and between countries) has now been included in the paper and is an important aspect of this paper and one worth highlighting perhaps more fully for this journal in terms of communication.

Reviewer #4:

[Was not available to re-review, however, in consultation with the other reviewers of this manuscript, we are satisfied that you have addressed Reviewer #4's concerns.]

Responses to the Reviewers' comments:

Reviewer #1 (Remarks to the Author):

Overall, this revised manuscript has addressed the concerns of the previous review. This reviewer continues to believe, however, that reporting ethnicity for the samples is important (even if flawed) to better understand the cohorts studied. Even if one does not believe that ethnicity has an impact on cognitive functioning with respect to genetic background, ethnicity in many situations can be a proxy for social and environmental issues that could impact cognitive development. I believe that readers of this manuscript will want that basic information. It is not necessary to relate ethnicity to the developmental outcome.

We appreciate the Reviewer's request. Therefore, we have included, in Supplementary Table 1, baseline socio-demographic data according to study site, which is the best proxy we can provide in these populations given that no information about ethnicity was collected. We have also added the following text to the Results section on p.6:

The baseline socio-demographic characteristics (Supplementary Table 1) of these samples were remarkably similar to those of the original cohorts at each study site,² and the samples across the five sites were, as expected, also similar because the same entry criteria were used. In addition, the satisfactory growth and development of the children at 1 and 2 years of age has confirmed the adequate health, educational and nutritional status of the original cohorts.⁹

Reviewer #3 (Remarks to the Author):

1) The issue of the psychometrics and validity of the developmental measurement tools is particularly important given the conclusions. There still appears to be limited interpretation and discussion of the paper in the context of what it is trying to achieve and the in the context of child development and measurement more generally. The authors have mentioned some of these implications in their responses to reviewers including implication for further use of their measurement approach in the context of developmental tools in low and middle income countries and some of the current limitations but have not included sufficient responses in the paper itself.

We agree with the Reviewer and so we have added to the manuscript some of our previous responses to the Reviewer's comments:

The psychometric properties of the measure we developed, the INTER-NDA, including internal consistency and construct validity, evaluation of its performance using interclass correlations for absolute agreement; Bland-Altman analyses for bias and limits of agreement, and sensitivity and specificity analyses for accuracy, have all been validated against the Bayley Scales of Infant Development.^{14,15} (Discussion, p.16)

We agree completely with this statement, especially in the light of the similarities in results obtained for anthropometric measures across populations when health and nutritional needs are met. International prescriptive standards for ECD are recommended for both clinical practice and research as many of the tools currently used are based on very skewed populations from developed countries, which little resemble present, culturally mixed societies. (Discussion, p.19)

Clearly, in clinical settings, it is important to focus on an individual child as the unit of diagnosis and interventions. At population level, the first step is to identify children at risk, i.e. those that require further assessment and are most likely to benefit from an intervention. However, the heterogeneity of

ECD measures presently in use and their reliance on specialists has made it difficult to carry out population-based ECD screening. (Discussion, p.19)

Lastly, with reference to the Reviewer's previous comment about the 3rd and 97th centiles, we have added the following text to the Methods section (p.22): **because they correspond closely to $\pm 2SD$, and they are the recommended cut-offs of the WHO Child Growth Standards, which are used internationally to evaluate children of this age.**

2) Similarly while the issue of deriving a theoretical population that is not representative makes sense and is alluded in the response comments this still needs further clarity in the paper itself as will seem odd to most reviewers.

We thank the Reviewer for making this important point. We have, therefore, added to the Results section (p.15) the following text, which was included in our initial response to the Reviewer:

The samples described are not representative of whole countries, regions or cities, nor were they intended to be. Instead, they were specifically selected - reflecting our conceptual approach to the issue - to include geographical areas populated by low-risk pregnant women and their children within each country or city. Hence, the samples are intended to represent a theoretical, healthy, low-risk population - rather than one that includes both low- and high-risk populations - within a country or city.

The issue of population variance (within and between countries) has now been included in the paper and is an important aspect of this paper and one worth highlighting perhaps more fully for this journal in terms of communication.

We thank the Reviewer for highlighting this crucial finding. For greater emphasis, as the Reviewer suggests, we have included the following sentence at the end of the Introduction (p.4):

We also show that the data can be pooled to create international standards because the variation in child development "between" sites is minimal compared to other sources of variance.